# BRITTLE INTERPRETATIONS: THE VULNERABILITY OF TCAV AND OTHER CONCEPT-BASED EXPLAINABILITY TOOLS TO ADVERSARIAL ATTACK

## ABSTRACT

Methods for model explainability have become increasingly critical for testing the fairness and soundness of deep learning. A number of explainability techniques have been developed which use a set of examples to represent a human-interpretable concept in a model's activations. In this work we show that these explainability methods can suffer the same vulnerability to adversarial attacks as the models they are meant to analyze. We demonstrate this phenomenon on two well-known concept-based approaches to the explainability of deep learning models: TCAV and faceted feature visualization. We show that by carefully perturbing the examples of the concept that is being investigated, we can radically change the output of the interpretability method, e.g. showing that stripes are not an important factor in identifying images of a zebra. Our work highlights the fact that in safety-critical applications, there is need for security around not only the machine learning pipeline but also the model interpretation process.

## 1 INTRODUCTION

Deep learning models have achieved superhuman performance in a range of activities from image recognition to complex games (LeCun et al., 2015; Silver et al., 2017). Unfortunately, these gains have come at the expense of model interpretability, with massive, overparametrized models being used to achieve state-of-the-art results. This is a salient problem when deep learning is applied to domains such as healthcare (Miotto et al., 2018), criminal justice (Li et al., 2018), and finance (Huang et al., 2020), where a prediction needs to be explainable to the user, leading to a surge of interest in tools that can illuminate the underlying decision making process of deep learning models.

Besides being inherently black-box in nature, deep learning models have also been shown to be vulnerable to adversarial attacks where small perturbations to model input result in dramatic changes to model output (Szegedy et al., 2013). This phenomenon is concerning when deep learning tools are deployed in safety-critical environments. A range of approaches have been developed to improve a model's robustness to adversarial attack (Silva & Najafirad, 2020), including the use of explainability methods to detect adversarial examples (Zhang et al., 2021; Wang & Gong, 2021). But if explainability methods are an important component in a machine learning system, then the robustness of these methods are nearly as important as the robustness of the model itself. In this paper we explore the vulnerability of concept-based interpretability methods (CBIMs). That is, methods that interrogate a model and its decisions based on a concept.

CBIMs usually rely on a user provided collection of positive examples (tokens) of a concept. While this flexibility makes these methods an attractive approach for understanding deep learning models in domains such as healthcare (Graziani et al., 2018b; Mincu et al., 2021), it also introduces a single point of failure wherein subtle changes to a few centralized tokens representing a concept results in misinterpretations of many subsequent input. We describe a threat model for post-hoc CBIMs, outlining adversary goals, knowledge, and capabilities. Then we introduce a family of attacks fitting this threat model which we call *token pushing (TP) attacks*. These learn small perturbations that when added to tokens of a concept result in remarkably different output for the interpretability method. Specifically, we optimize our perturbations so that when they are added to a token, they significantly change a model's internal representation of the input.

We test TP attacks against two popular CBIMs: Testing with Concept Activation Vectors (TCAV) (Kim et al., 2018) and Faceted Feature Visualization (FFV) (Goh et al., 2021). While TCAV and FFV are similar in that they are both concept-based, their output is quite different. TCAV quantifies the extent to which a concept is important to a model's prediction for a specific input dataset. A variant of this method was recently a central component of (McGrath et al., 2021), which used it to provide evidence that models such as AlphaZero learn human chess concepts. FFV on the other hand, can be used to produce visualizations that represent how individual neurons capture a specific concept. We show that TP attacks are effective for both TCAV and FFV. For example, a TP attack causes TCAV to give output indicating that stripes are not an important feature to the class 'zebra.' On the other hand, a TP attack can radically change the feature visualizations generated by FFV (Figure 3).

We evaluate TP attacks on pretrained ImageNet models (Deng et al., 2009; Marcel & Rodriguez, 2010) using the Describable Textures Dataset (Cimpoi et al., 2014) for concept tokens. Through our experiments we show that a TP attack does not require the adversary to know what interpretability method is being used. The same perturbations that cause TCAV to fail, also cause FFV to fail. Finally TP attack possesses moderate transferability, meaning that as long as a surrogate model is available, it can be applied even when the defender model architecture is unknown.

In summary, our contributions in this paper include the following.

- Formalization of an adversarial threat model for post-hoc concept-based interpretability methods that identifies concept tokens as a single point of failure for such methods.
- Introduction of TP attacks which cause misinterpretation by disrupting a central mechanism by which concept tokens are used across a range of concept-based interpretability methods.
- Demonstration of the effectiveness of TP attacks on two concept-based interpretability methods, TCAV and FFV.
- Introduction of the first (to our knowledge) adversarial attack on feature visualization.

## 2 RELATED WORK AND BACKGROUND

**Interpretability methods:** Because of the size and complexity of modern deep learning architectures, skill is required to extract interpretations of how these models make decisions. Established methods range from those that focus on highlighting the importance of individual input features to those that can give clues to the importance of specific neurons to a particular class. Popular examples of interpretability methods that focus on input feature importance include saliency map methods (Selvaraju et al., 2017; Sundararajan et al., 2017; Ribeiro et al., 2016; Fong & Vedaldi, 2017; Dabkowski & Gal, 2017; Chang et al., 2019) which identify those input features (for example, pixels in an image) whose change is most likely to change the network's prediction.

CBIMs focus on decomposing the hidden layers of deep neural networks with respect to human-understandable concepts. One of the best-known approaches in this direction involves the use of concept activation vectors (CAVs) (Kim et al., 2018) which we describe in detail in the next section. Work that is either related or extends these ideas includes (Zhou et al., 2018; Graziani et al., 2018a; 2019; Koh et al., 2020; Yeh et al., 2020).

Feature visualization is a set of interpretability techniques (Szegedy et al., 2014; Mahendran & Vedaldi, 2015; Wei et al., 2015; Nguyen et al., 2016b) concerned with optimizing model input so that it activates some specific node or set of nodes within the network. However, a challenge arises when one tries to analyze 'polysemantic neurons' (Olah et al., 2018), neurons that activate for several conceptually distinct ideas. For example, a neuron that fires for both a boat and a cat leg is polysemantic. Interpretability methods have imposed priors to disambiguate neurons by clustering the training images (Wei et al., 2015; Nguyen et al., 2016b) or the hidden layer activations (Carter et al., 2019) and using the average of the cluster as a coarse-grained image prior, parameterizing the feature visualization image with a learned GAN (Nguyen et al., 2016a), or using a diversity term in the feature visualization objective (Wei et al., 2015; Olah et al., 2017).

**Robustness of interpretability methods:** This is not the first work that has shown that interpretability methods can be brittle. Saliency methods have been shown to produce output maps that appear to point to semantically meaningful content even when they are extracted from untrained models, indicating that these methods may sometimes simply function as edge detectors (Adebayo et al.,

2018). Further, preliminary work has studied the robustness of Concept Bottleneck Models, an intrinsically interpretable concept-based method, to out-of-distribution data (Koh et al., 2020). From a more adversarial perspective, a number of works have shown that saliency methods are vulnerable to small perturbations made to either an input image or to the model itself that cause the model to offer radically different interpretations (Heo et al., 2019; Ghorbani et al., 2019; Viering et al., 2019; Subramanya et al., 2019; Anders et al., 2020); work has looked at methods to make explanations more robust to attack (Lakkaraju et al., 2020). On the other hand, this is the first work that shows that CBIMs are also vulnerable to adversarial attack. In particular, since we focus on attacks targeting a component absent from other interpretability methods (concept tokens), there is not a straightforward way of applying the attacks mentioned above within the threat model presented in this paper.

## 2.1 TCAV AND LINEAR INTERPRETABILITY

In this section we describe the method of testing with concept activation vectors (TCAV) (Kim et al., 2018). Let $f : X \to \mathbb{R}^d$ be a neural network which is composed of $n$ layers and designed for the task of classifying whether a given input $x \in X$ belongs to one of $d$ different classes. Write $f_\ell : X \to \mathbb{R}^{d_\ell}$ for the composition of the first $\ell$ layers so that $f_n = f$ and $d_n = d$ and let $h_\ell : \mathbb{R}^{d_\ell} \to \mathbb{R}^d$ be the composition of the last $n - \ell$ layers of the network so that $f = h_\ell \circ f_\ell$ for any $1 \le \ell \le n - 1$. Let $C$ be a concept for which we have a set of positive examples (tokens) $P_C = \{x_i^P\}_i$ and negative examples $N_C = \{x_i^N\}_i$, both belonging to $X$. These are represented in the $\ell$th layer of $f$ as the points $f_\ell(P_C)$ and $f_\ell(N_C)$ respectively. One can apply a binary linear classifier to separate these two sets of points, resulting in a hyperplane in $\mathbb{R}^{d_\ell}$. This hyperplane can be represented by two unit normal vectors. We choose the one, $v_C^\ell \in \mathbb{R}^{d_\ell}$, that points into the region corresponding to the points $f_\ell(P_C)$. $v_C^\ell$ is called the *concept activation vector* in layer $\ell$ associated with concept $C$. One can think of $v_C^\ell$ as the vector that points toward $C$-ness in the $\ell$th layer of the network.

Let $h_{\ell,k}$ denote the $k$th output coordinate of $h_\ell$ corresponding to class $k$. In the classification setting, $h_{\ell,k}$ then represents the model's confidence that input belongs to class $k$. To better understand the extent to which concept $C$ influences the model's confidence of $x \in X$ belonging to class $k$ we compute:

$$S_{C,k,l} = \nabla h_{\ell,k} \left( f_\ell(x) \right) \cdot v_C^l. \tag{1}$$

A positive value of $S_{C,k,l}$ indicates that increasing $C$-ness of $x$ makes the model more confident that $x$ belongs to class $k$. The *magnitude TCAV score* for a dataset $D$ is defined as

$$\text{TCAV}_{Q_{C,k,\ell}} = \frac{1}{|D_k|} \sum_{x \in D_k} S_{C,k,l}(x),$$

where $D_k$ is the subset of $D$ consisting of all instances predicted as belonging to class $k$. We compare the TCAV magnitude of the positive concept images with the TCAV magnitude for random images in the layer, and use a standard two-sided $t$-test to test for significance. We can also compute the *relative TCAV score*, which replaces the set of negative natural images in $N_C$ with images representing a specific concept.

## 2.2 FACETED FEATURE VISUALIZATION

Goh et al. (2021) introduced a new concept-based feature visualization objective for neuron-level interpretability, *Faceted Feature Visualization (FFV)*. The objective disambiguates poly-semantic neurons by imposing a prior towards a linear concept $C$ in the optimization objective. Goh also utilizes a set of positive and negative examples of a concept $C$ ($P_C$ and $N_C$ respectively). Similar to the TCAV method, one trains a binary linear classifier on the image of these two sets under the map $f_\ell$ to obtain $v_C^l$. To visualize output that tends to activate a neuron at layer $\ell$, position $i$, while at the same time steering the visualization toward a specific context, the authors solve the following optimization problem:

$$\arg \max_{x \in X} f_{\ell,i}(x) + v_C^l \cdot (f_\ell(x) \odot \nabla f_{\ell,i}(x)), \tag{2}$$

where $\odot$ is the Hadamard product. Note that the first term helps find $x$ which result in a strong activation of $f_{\ell,i}$, while the second term finds $x$ such that $f_\ell(x)$ tends to point in the direction of $v_C^\ell$.

## 3 Adversarial attacks on interpretability

An adversarial attack (Szegedy et al., 2013) on a model $f$ is a small perturbation $\delta$ that, when applied to a specific input $x$, results in large changes to model prediction $f(x)$. The meaning of 'small' is usually specified by a metric such as an $\ell_p$-norm and can either be a hard or soft constraint. In this work we use projected gradient descent (PGD) (Madry et al., 2018) to construct our attacks, but this should be seen mostly as a placeholder. The novelty of the attack is the manner in which it targets the underlying mechanism central to many CBIMs. Optimization approaches other than PGD could doubtless be used for the same effect.

We frame the notion of a CBIM abstractly in order to better understand its attack surface. We view such a method as a map that takes (1) a model, (2) positive tokens of the concept that we would like to steer our interpretation, (3) negative tokens of the concept and (4) an *interpretation target* which will be the focus of the interpretation. We call the output of an interpretability method an *interpretation object*. An interpretation object might be a single scalar value (as in the case of TCAV), or it may be an image (as in the case of FFV). In all cases, an interpretation object is designed to help the user better understand a model's decision making process. Thus, we can understand an interpretability method as a function $I : \mathcal{M} \times \mathcal{P} \times \mathcal{N} \times \mathcal{T} \to \mathcal{O}$, where $\mathcal{M}$ is the collection of models that can be interpreted, $\mathcal{P}$ is the space of all possible positive token sets, $\mathcal{N}$ is the space of all possible negative token sets, $\mathcal{T}$ is the space of interpretation targets, and $\mathcal{O}$ is the space of interpretation objects that the method produces. We note that in the case of TCAV, the interpretation target is a dataset $D_k$ of examples of some class $k$, while the interpretation target of FFV is a specific node position $(i, j, k)$ in the model.

### 3.1 A threat model for attacks on concept-based interpretability methods

Following a suggestion given in (Carlini et al., 2019), we state the threat model that we will consider in this paper. Since we will only be considering images as input in our experiments, we specify to that setting here. Otherwise, we use the formalism that we developed above. Specifically, we assume there exists an interpretability method $I$, a model $f \in \mathcal{M}$, set of positive image tokens $P_C = \{x_i^P\} \in \mathcal{P}$, set of negative image tokens $N_C \in \mathcal{N}$, and interpretation target $T \in \mathcal{T}$. We also assume a function $F : \mathcal{O} \times \mathcal{O} \to \mathbb{R}$ that quantitatively captures meaningful difference between interpretation objects.

**Adversary's goal:** The adversary's goal is to find perturbations $\{\delta_i\}_i$ such that $\hat{P}_C = \{x_i^P + \delta_i\}$ maximizes the value of $F(I(f, P_C, N_C, T), I(f, \hat{P}_C, N_C, T))$. That is, the change from $P_C$ to $\hat{P}_C$ maximizes the difference in interpretation as measured by $F$. In order to avoid detection, $\hat{P}_C$ is subject to the constraint: $\max_i ||\delta_i||_\infty \leq \epsilon$, for some fixed $\epsilon > 0$.

**Adversary knowledge and capabilities:** (1) In this paper we assume that the adversary has read and write access to the tokens $P_C$ either before or after they have been collected. (2) We do not assume that the adversary has access to either $T$ (the dataset of examples predicted as belonging to class $k$ in the case of TCAV or the specific neuron position that is being targeted in the case of FFV). We do assume that the adversary knows the hidden layer that is being targeted for both TCAV and FFV. (3) We assume that the adversary has read access to at least a surrogate model trained on the same dataset as $f$. We do not assume that this surrogate model needs to have the same architecture as $f$. (4) Finally, we do not assume that the adversary knows the interpretability method that will be used.

The adversary's goal is framed in terms of a function $F$ that depends on the specific interpretability method. This might seem to be in conflict with assumption (4) that says that the adversary does not have knowledge of the interpretability method being used. Actually, we show that TP attacks, which we propose below, work for $F$ specific to both TCAV and FFV simultaneously by optimizing for an objective function that disrupts the fundamental mechanism by which TCAV, FFV, and other CBIMs work. As noted in the introduction, we centered our threat model around the positive tokens critical to CBIMs that, once perturbed, can cause persistent misinterpretation across numerous inputs. In contrast, a perturbation of an individual input image alone affects only the interpretation associated to that input.

Figure 1: A schematic of the TP attack. $P_C$ is the original set of positive examples of concept $C$, $N_C$ is the set of negative examples of concept $C$, $U_C$ are the unrelated examples that are used to calculate $\mu_T$, and $\hat{P}_C$ is the set of positive examples after the attack. Note that positive examples get pulled toward unrelated concept example centroid $\mu_T$, changing the direction of $v_C^\ell$.

## 3.2 ATTACKING TOKENS OF A CONCEPT

In this section we introduce the *token pushing (TP) attack*. The basic idea is simple; we find perturbations $D_C = \{\delta_i\}_i$ that significantly alter a model's internal representation of the concept tokens $P_C = \{x_i^P\}_i$. Using the notation developed in 3.1, we assume that the adversary has access to a copy of the defender's model (or a surrogate model) $f : X \to \mathbb{R}^d$, the hidden layer that the interpretation method will use, and write access to the set of tokens $P_C$ that represent a concept $C$.

The perturbations added to each element in $P_C$ shifts its hidden representation in layer $\ell$ so that it no longer correlates with concept $C$. In order to find a point that can guide this shift, the first step is for the adversary to choose some collection of images that are unrelated to $C$, $U_C := \{x_i^U\}_i$. The adversary calculates the centroid of $f_\ell(U_C)$, which we denote by $\mu_T$, which will serve as a representative of "unrelatedness" to $C$. Then for each $x_i^P \in P_C$, the adversary uses PGD to compute

$$\delta_i := \underset{\|\delta\|_\infty \leq \epsilon}{\arg\min} ||f_l(x_i^P + \delta) - \mu_T||. \tag{3}$$

This is related to the hidden layer attacks described in (Wang et al., 2018; Inkawhich et al., 2019). A schematic of the TP attack can be found in Figure 1. Examples of the perturbations can be found in Figure 7 in the Appendix.

In Section 4, we show that in spite of the fact that Equation 3 is neither the interpretability objective of TCAV nor FFV, it is still effective when applied to either method. In fact, objective function 3 makes the TP attack more flexible since it acts against the underlying mechanism common to both these and other interpretability methods: the spatial proximity of hidden representations of input that are semantically related. This means that the adversary does not need to know the specific interpretability method that the defender is using. This also means that the attacker does not need access to the interpretation target, as they would if they were to optimize against the interpretability objective directly.

## 4 EXPERIMENTS

To better understand the effectiveness of the methods proposed in Section 3.2, we apply our attacks to TCAV and FFV in the case that they are used to interrogate an InceptionV1 model (Szegedy et al., 2015) that has been trained on ImageNet-1k (Deng et al., 2009). We choose InceptionV1 because it

| | InceptionV1 Layer | | | |
|---|---|---|---|---|
| **Attacks** | mixed3a | mixed3b | mixed4a | mixed4b |
| Baseline TCAV (no attack) | $0.69 \pm 0.02$ | $0.90 \pm 0.01$ | $0.66 \pm 0.03$ | $0.68 \pm 0.04$ |
| Gaussian noise | $0.61 \pm 0.02$ | $0.62 \pm 0.02$ | $0.64 \pm 0.03$ | $0.67 \pm 0.04$ |
| *PGD attack on* | | | | |
| Logit | $0.37 \pm 0.02$ | $0.37 \pm 0.03$ | $0.35 \pm 0.02$ | $0.33 \pm 0.03$ |
| mixed3a centroid | $\mathbf{0.29 \pm 0.05}$ | $0.29 \pm 0.10$ | $0.22 \pm 0.05$ | $0.34 \pm 0.08$ |
| mixed3b centroid | $0.17 \pm 0.05$ | $\mathbf{0.39 \pm 0.10}$ | $0.19 \pm 0.03$ | $0.37 \pm 0.08$ |
| mixed4a centroid | $0.22 \pm 0.06$ | $0.40 \pm 0.11$ | $\mathbf{0.32 \pm 0.05}$ | $0.44 \pm 0.08$ |
| mixed4b centroid | $0.27 \pm 0.07$ | $0.32 \pm 0.10$ | $0.33 \pm 0.06$ | $\mathbf{0.42 \pm 0.08}$ |
| mixed4c centroid | $0.26 \pm 0.08$ | $0.30 \pm 0.09$ | $0.29 \pm 0.05$ | $0.28 \pm 0.08$ |
| mixed4d centroid | $0.28 \pm 0.08$ | $0.30 \pm 0.10$ | $0.25 \pm 0.06$ | $0.18 \pm 0.10$ |

Table 1: The TCAV magnitude score for the zebra class on the 'striped' concept, before and after the TP attacks on InceptionV1. The Baseline TCAV row uses the concept sets with no perturbations. The rows below 'PGD attack on' indicate the layer that is being targeted by the TP attack. The columns are the InceptionV1 layer that TCAV is being applied to. For all concept/pairs, We bold those values where the layer targeted by the TP attack and the layer TCAV is applied to are the same.

is a model commonly used in the interpretability literature (Kim et al., 2018; Olah et al., 2020) and choose ImageNet-1k since it is easy to obtain high-quality weights for this model/dataset combination. The token sets that we used to capture concepts come from the Describable Textures Dataset (DTD) (Cimpoi et al., 2014). We perform all PGD attacks with $\epsilon = 4/255$ and 20 steps. For the CAV, we use a linear classifier trained via stochastic gradient descent and $\ell_2$-regularization. See the Appendix for more experiment details.

To test the TP attack against TCAV, we choose concept/class pairs with straightforward associations: 'stripes'/'zebra', 'honeycombed'/'honeycomb', as well as the 'scaly' concept with four separate snake classes: 'Green snake', 'Hognose snake', 'Water snake', or 'King snake'. We perform the same experiment for all concept/class pairs, but for simplicity explain the procedure with the 'stripes'/'zebra' concept/class pair. We select 70 sets of 50 randomly chosen images from ImageNet $\{N_C^i\}$ which do not intersect. The same $\{N_C^i\}$ will be used for all concept/class pairs. We also fix a set of unrelated images $U_C$ of size 1000 that are also randomly sampled from ImageNet. Finally, we choose random sets of 40 images from the classes 'stripes' $P_{\text{striped}}$ from DTD. $D_k$ is a collection of images which the InceptionV1 model predicts as 'zebra'.

For each layer of InceptionV1 we run the TP attack against $P_{\text{striped}}$. For each of the resulting pairs $(P_{\text{striped}}, \hat{P}_{\text{striped}})$ and each layer of InceptionV1, we then apply TCAV 70 times (once for each $N_C^i$), calculating the difference in magnitude TCAV score between $P_{\text{striped}}$ and $\hat{P}_{\text{striped}}$. Numerical results for 'stripes'/'zebra' can be found in Table 1. The smaller values indicate a change in magnitude TCAV score before and after the attack. Plots of the raw TCAV magnitude scores for both the clean positive tokens and the attacked positive tokens (where each attack targeted a different layer of InceptionV1) are found for the 'scaly'/'snake' concept pairs in Figure 2. Sample concept images before and after the attack, as well as CAVs visualized with empirical DeepDream (Mordvintsev et al., 2015) before and after the attack can be found in the appendix (notably, we find that besides some changes in coloring, DeepDream still produces visualizations that resemble the original concept, even when it is applied to the perturbed tokens, providing another type of imperceptibility of this attack).

We include 95% confidence intervals for each layer based on the 70 different $N_C^i$ sets. The point of this is to verify that the result does not depend on having the "right" negative examples. To test that the attack perturbations work for reasons other than the fact that they are perturbations, we also apply TCAV to positive token sets to which we have added random Gaussian noise with the per-channel mean and standard deviation of the PGD logit attack. Finally, note that we also include the results showing what happens when a TP attack targets a different hidden layer than TCAV is being applied to (these are in the off-diagonal of Table 1). A version of our results for relative TCAV can be found in Appendix A.2.

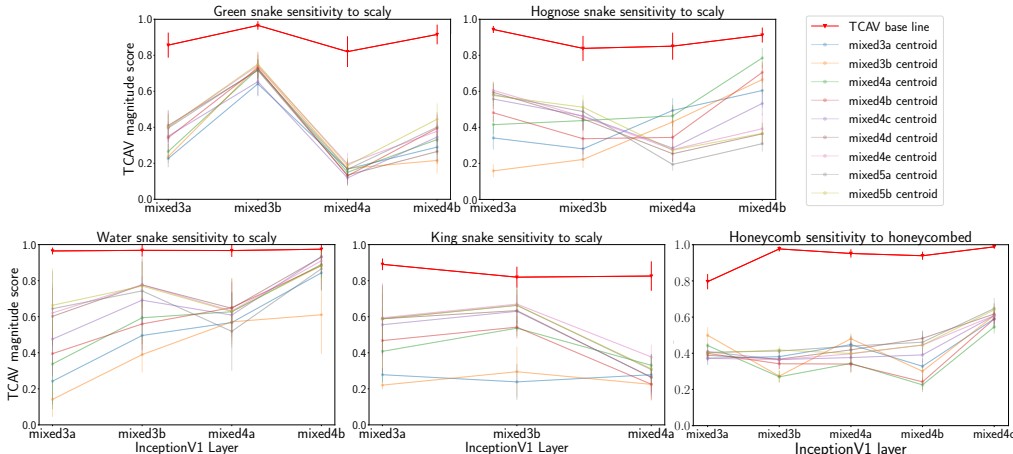

Figure 2: Adversarial attacks on the 'scaly' concept set for four snake classes in ImageNet-1k, as well as the 'honeycombed' concept set for the honeycomb class. Each curve represents a TP attack targeting a different layer of InceptionV1. We use the same set of perturbations across each snake class. The $x$-axis records different layers of the InceptionV1 network, restricted to layers where the snake class is sensitive to the un-modified scaly concepts, according to the TCAV sign score. The $y$-axis is the TCAV magnitude score when TCAV is applied to that layer.

Figure 3: A feature visualization without any concept prior (first row), faceted feature visualization on the same neuron for 'stripes', 'dots', 'zig-zags' (second row), and the faceted feature visualization after it has been attacked (third row) of channel 9 on InceptionV1 layer mixed4d. While visualizations in the second row reflect the concept prior, the visualizations in the third row do not (indicating the attack was successful). An example token perturbation for this attack is found in Figure 7.

We evaluate the token perturbation attack on FFV by performing feature visualizations for InceptionV1 on every channel neuron for the layers mixed3a, mixed3b, mixed4a, and mixed4b. We use the feature visualization objective equation 2, and compare feature visualizations with clean concept images $P_C$, concept images with Gaussian noise, and a concept set with perturbations created by PGD on the respective hidden layer with equation 3. We give an example of FFV output before and after an attack in layer mixed4d in Figure 3.

We quantitatively test the effectiveness of the attack on FFV by using a variant of the Fréchet Inception Distance (FID) (Heusel et al., 2017) to measure the perceptual distance between feature visualizations. Namely, we compare feature visualizations created with the channel objective (i.e., using only the

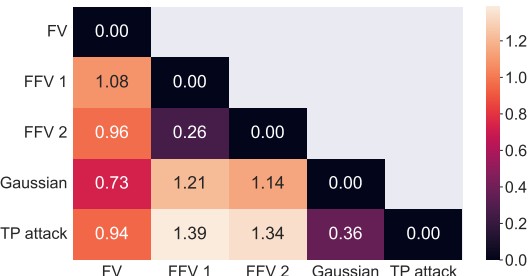

Figure 4: Fréchet Inception Distance scores for feature visualizations on the layers mixed3a, mixed3b, mixed4a, and mixed4b. Five different feature visualizations are performed: channel feature visualization (FV), two separate runs of Faceted Feature Visualization with different sets of positive and negative concept images (FFV 1 and FFV2), with Gaussian noise added to the positive concept images (Gaussian), and with the token pushing attack (TP attack).

first term in equation 2), FFVs created with two sets of random images $P_C$ and with clean stripe concepts $P_C$, FFVs with a set of stripe concepts $P_C$ with a perturbation created via targeting the layer mixed3b with equation 3, and FFVs where we add Gaussian noise to $P_C$. The FID score is calculated across layers for every channel neuron in InceptionV1 layers: mixed3a (256 channels), mixed3b (480 channels), mixed4a (512 channels), and mixed4b (512 channels), shown in Figure 4. We use a PyTorch implementation of FID (Seitzer, 2020) and use the second block of InceptionV3 as the visual similarity encoder (due to the smaller dataset size).

## 5 RESULTS

Our results show that TP attacks effectively change the output of both TCAV and FFV from the baseline interpretability results. For TCAV, we can consistently lower the TCAV magnitude score that indicates the relative importance of a concept to an output class. In Table 1, we measure the TCAV magnitude score on four early layers of InceptionV1. For each run and layer, we take the average difference between the TCAV magnitude score for the striped concept set and a random concept set over 70 sets of random images. We note that, unsurprisingly, attack success tends to increase when the layer that an attack was developed for and the layer TCAV is being applied to are the same. However, we also find that the attack remains effective even when these are not the same. For example, in Table 1, the attack targeting the layer 'mixed4b' is successful across all layers examined. We also observe this in Figure 2, where all attacks effectively modify the TCAV magnitudes on the 'scaly' concept for the 'snake' classes for all of the layers examined.

For FFV, we can observe the TP attack effectiveness from the visual differences between 1) a channel feature visualization (i.e., a feature visualization that optimizes the first term in equation 2), 2) the faceted feature visualization with a clean concept set $P_C$, and 3) the faceted feature visualization with a perturbed concept set $\hat{P}_C$. We give three such examples separately using the striped, dotted, and zig-zagged concept sets in Figure 3. We use FID as a measure of visual difference, and test the effectiveness of the TP attack on FFV for the 1,760 channel neurons in the InceptionV1 layers 'mixed3a', 'mixed3b', 'mixed4a', and 'mixed4b'. We use the striped concept set and perform two separate FFV visualizations for each neuron using different sets of negative concept set images. Figure 4 shows that the FID scores between the separate clean FFV runs is $0.26$, while the FID score between the TP attack and the clean FFV runs are $1.39$ and $1.34$. The significantly larger FID scores suggest that the TP attack modifies the FFV output more than the variation between runs. This, along with visualizations such as 3, suggest that a TP attack can drastically change the semantic meaning associated with the feature visualizations produced by FFV.

Finally, we find that both the TCAV magnitudes (Table 1) and the FFV FID scores (Figure 4) are susceptible to Gaussian noise added to the concept set. This suggests that, even independent of adversarial attacks, CBIMs are brittle. This brittleness suggests that these methods are also vulnerable to natural distribution shifts in data, e.g., between the concept set and training images. We see a need for research into robust interpretability methods.

Figure 5: TCAV sensitivity scores for the zebra class with the stripe images for a ResNet-18 (left) (He et al., 2016) and MobileNetV2 (right) (Sandler et al., 2018) trained on ImageNet-1K. The attacks uses perturbations made on the stripe concept images for InceptionV1 using centroids for different hidden layers. All layers/blocks shown are sensitive to the stripe concept before the attack, and are not sensitive after the attack.

## 5.1 TRANSFERRABILITY

As noted in 3.1, the knowledge required for an adversary to implement an attack is decreased significantly if they do not need to know the specific model being used by the defender. We therefore test the transferrability of the TP attack by applying TCAV to other model architectures trained on ImageNet: a ResNet-18 and MobileNetV2. We again consider the concept/class pair stripes/zebra with the same set of $N_{stripes}$, $U_{stripes}$, and $P_{stripes}$ that were used for Table 1. We compute the TCAV magnitude score for stripes/zebra for the four residual blocks in ResNet-18 and the three layers in MobileNetV2 that were sensitive to the stripe concept according to signed TCAV. Figure 5 compares a baseline score with scores for TP attacks applied to the different layers.

We find that the TP attacks targeting any of the 7 layers of InceptionV1 result in significant decreases in TCAV magnitude score when applied to the first block of ResNet18 and all three layers of the MobileNetV2. The transfer TP attack does not seem to be effective against Block 2 and Block 4 of the Resnet-18. These results point toward TP attack being moderately transferable, especially when TCAV is being applied to earlier layers of the defender's model.

## 5.2 LIMITATIONS

In this work we chose two CBIMs to test TP attacks on. While TCAV and FFV capture some of the diversity of such methods, they do not capture their full breadth. In particular, it would be useful to understand how TP attacks behave when they are applied to other types of feature visualization methods, namely those that average over a large number of images or activations (Nguyen et al., 2016b; Carter et al., 2019) to build a visualization. Further, while we only consider image classification models, TCAV is agnostic to modality. Evaluating interpretability method brittleness in other critical modalities such as NLP would give a more complete picture of these method's vulnerabilities. Finally, we focus on perturbations to positive concept tokens. To fully understand the attack surfaces of CBIMs, it makes sense to consider attacks on the other inputs to a method: the model itself, negative examples, and the interpretation targets. As a limited example, an adversarial attack may be designed to be 'triggered' for only certain concept and dataset interpretation combinations.

## 6 CONCLUSION

In this work we show that concept-based interpretability methods, like much of the deep learning modeling pipeline, are vulnerable to adversarial attacks. By subtly changing the examples of a concept that a user wishes to use to interrogate a model, an adversary can induce radically different interpretations. The attacks we describe are general enough that they work for multiple interpretability methods without modification (FFV and TCAV). We hope that the results of this paper will promote better security practices, not only around the model pipeline itself, but also around the method that is being used to interpret the model.

## 7    REPRODUCIBILITY STATEMENT

In the interest of making our results reproducible and able to be easily expanded upon, we make our codebase available to the public, including our implementations of the centroid PGD attack and the faceted feature visualization we used. We also include attack and evaluation scripts with sensible defaults and examples. Finally, we provide the data used throughout this paper, including our feature visualizations. This entire repository will be available on a public GitHub repository once the anonymous review period has completed.

## 8    ETHICS STATEMENT

In this work we highlight the vulnerability of a class of popular interpretability methods to adversarial attack. We chose to explore a threat model wherein the positive tokens for a concept are perturbed. This is of particular concern because (unlike individual input) positive tokens will often be centralized and used collectively by researchers and practitioners many times. Because of this, an attack on a single data source may have wide-ranging effects. We hope that by better understanding and communicating this specific threat to interpretability, we can motivate researchers to use best practices around security for interpretability and explainability as they are already encouraged to do for dataset and model creation.

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

# A APPENDIX

## A.1 EXPERIMENT DETAILS

We use InceptionV1, ResNet-18, and MobileNetV2 models with pretrained weights from Torchvision (Marcel & Rodriguez, 2010). To run TCAV, FFV, and our attacks, we use PyTorch with an NVIDIA Tesla T4 GPU provided with Google Colab Pro as well as a single NVIDIA Tesla P100 GPU. We use the Captum (Kokhlikyan et al., 2020) implementation of TCAV with a linear classifier trained via stochastic gradient descent and $\ell_2$-regularization.

For the Faceted Feature Visualizations, we start with random noise and parameterize the image Fourier basis (Olah et al., 2017). We use random scaling, rotation, color, and shift transformations.

Figure 6: Relative TCAV magnitude scores before (top) and after (bottom) the PGD logit attack on the striped concept images for the striped, zig-zagged, and dotted concept sets.

## A.2 TP ATTACK ON RELATIVE TCAV

Here, we give an example experiment showing that TP attacks are also effective for a variant of TCAV, using relative TCAV scores. The results in Figure 6 use concept sets for stripes, ziz-zags, and polka-dots of 35 images each. Perturbations are made on the striped concept set using the final logit layer, towards an unrelated class (the $999^{th}$ 'toilet tissue' ImageNet-1k class).

## A.3 EMPIRICAL DEEPDREAM WITH THE CAVS

We use empirical DeepDream in Figure 8 to visualize the effect of the hidden layer PGD attack on the Concept Activation Vectors (Mordvintsev et al., 2015; Kim et al., 2018). We consider CAVs for the hidden layers mixed3b and mixed4b of InceptionV1. We use images from the striped, honeycombed, and scaly concept sets, and use perturbations found via Projected Gradient Descent attack on the hidden layer mixed4d. We use cosine similarity (Carter et al., 2019) for the feature visualization objective. We use the same Fourier parameterization and transformations we used for the Faceted Feature Visualization.

We note that the visualizations for the attacked CAV tend to qualitatively resemble those of the CAV without the attack, albeit with unnatural hue and colors. It has been proposed that DeepDream can confirm that CAVs represent the concept of images (Kim et al., 2018). Given the success of our adversarial attack, using DeepDream as a qualitative concept check for a CAV may therefore be misleading and provides evidence for the imperceptibility of the token pushing attack.

Figure 7: Example of stripe concept images before (left) and after (right) a TP attack. We use $\epsilon = 4/255$ and 20 iterations for all PGD experiments. The perturbation shown targets InceptionV1 layer mixed3a.

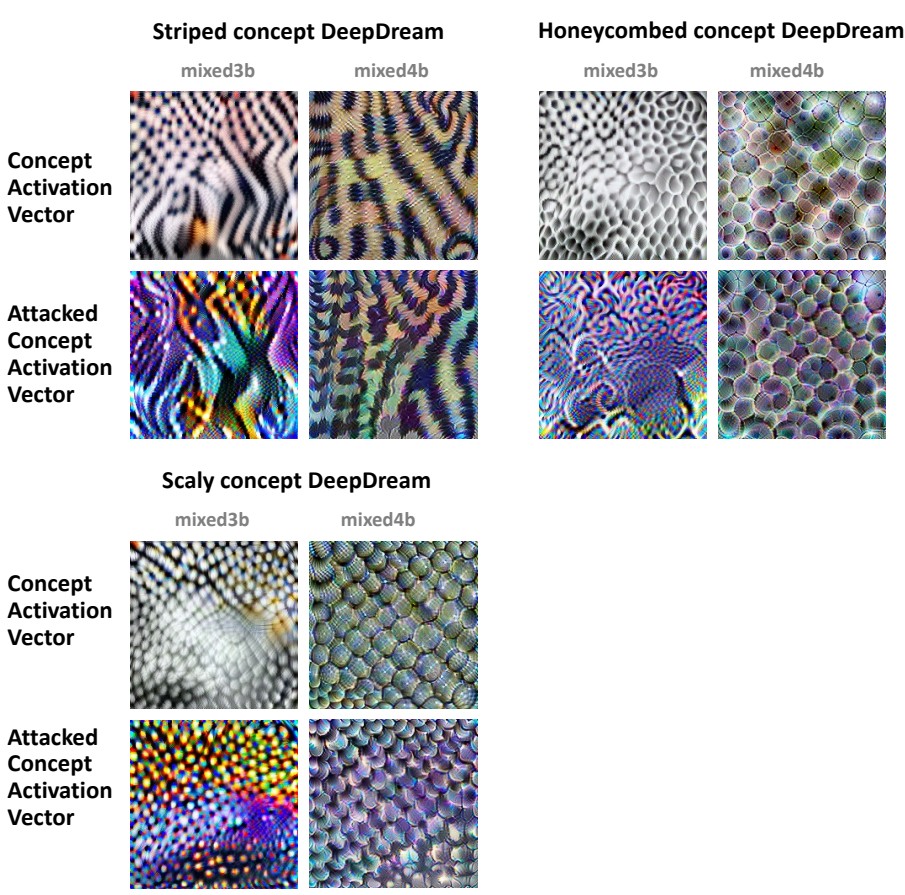

Figure 8: Empirical Deepdream Mordvintsev et al. (2015) with the Concept Activation Vectors (CAVs) computed with 1) the normal concept sets in the 'Concept Activation Vector' rows and 2) the perturbed concept sets in the 'Attacked Concept Activation Vector' rows. For the attacked concept sets, we use the PGD attack performed on the hidden layer mixed4d.

