# OpenReview forum: "Brittle interpretations: The Vulnerability of TCAV and Other Concept-based Explainability Tools to Adversarial Attack"
_ICLR.cc/2022/Conference — ICLR 2022 Submitted_

### Official Review · Reviewer_jcjC · 2021-10-24

**Correctness:** 3
**Technical Novelty And Significance:** 3
**Empirical Novelty And Significance:** 2
**Recommendation:** 5
**Confidence:** 4

**Main Review:**

### Strengths
The paper tackles a novel problem of adversarial robustness of concept-based explanations, and the proposed token pushing attacks could be of interest to the community. The proposed method is simple, and the paper is generally clear and well-written.

---

### Weaknesses:

- The motivation of the proposed method is unclear. The paper perturbs the hidden representations to demonstrate the concept-based explanations' vulnerability; however, the injected perturbation is not imperceptible. For instance, the attacked FFV in Figure 3 clearly shows the presence of the perturbation, which limits the practicality of the proposed attack framework. Therefore, I suggest evaluating with lower $\epsilon$ and $\ell_2$ or $\ell_1$ norm to make the perturbation sparse and imperceptible.
- In light of the above concern, existing adversarial detectors [1,2] can defend against the proposed attack. Can the authors validate their proposed attack against these adversarial detectors?
- Further, it would have been more effective to inject imperceptible perturbation in the input space to show that the concept-based explanations are vulnerable, as concept-based explanations are comparatively more robust to adversarial attacks in the input space (see Appendix A [3])
- The experimental evaluation is limited to two concept/class pairs on a single dataset, making it difficult to conclude the effectiveness of the proposed method. Therefore, I recommend evaluating multiple datasets (e.g., OAI or CUB datasets have concept attributes) and recent concept-based methods [4,5].


***Other questions***
- What is the significance of negative values in Table 1? Instead of reporting the change in magnitude, it might be more useful to evaluate the change in prediction accuracy after the attack on concepts following prior works [5].
- The results for transferability are intriguing; can the authors report the effect of transferability on different architectures?
- Can the authors show the visualization after the attack on the relative TCAV shown in Figure 6? Does the visualization resemble the dotted concept after the attack?

---

### References
[1] Roth and Khilcher et al. The Odds are Odd: A Statistical Test for Detecting Adversarial Examples. ICML 2019.
[2] Yang et al. ML-LOO: Detecting Adversarial Examples with Feature Attribution. AAAI 2020.
[3] Kim et al. Interpretability Beyond Feature Attribution: Quantitative Testing with Concept Activation Vectors (TCAV). ICML 2018.
[4] Yeh et al. On Completeness-aware Concept-Based Explanations in Deep Neural Networks. NeurIPS 2020.
[5] Koh et al. Concept Bottleneck Models. ICML 2021.

**Summary Of The Paper:**

The paper describes the threat model for concept-based interpretability methods and mainly investigates the vulnerability of TCAV and faceted feature visualization. It introduces Token Pushing (TP) attacks, which learn small perturbations for the token of concept leading to different output for the interpretability method. The proposed attack is evaluated on pre-trained ImageNet models on the Describabke Texture Dataset for concept tokens.

**Summary Of The Review:**

In the current form, I recommend rejection due to the above-mentioned reasons. In particular, the paper needs to clearly justify the motivation of the work with a stronger experimental evaluation to demonstrate the effectiveness of the proposed method.

---

> ### Author Response · Authors · 2021-11-23
> **Response to jcjC**
>
> We would like to start out by thanking the reviewer for taking the time to read our paper.
>
> - The reviewer states that the attack we propose is not imperceptible based on Figure 3. Figure 3 shows the output of FFV before the model has been attacked (first row) and after the model has been attacked (second). For the attack to be successful, the changes to FFV output should be significant (this is the adversary’s goal) because this means that the defender will be let into misinterpretation. In the first row, we see that when the unperturbed sets of tokens are used, we get feature visualizations that correspond to the concepts. We see in the second row that when we use the perturbed tokens, we get feature visualizations that do not resemble the associated concepts. On the other hand, this figure does not show an example of the perturbed positive tokens, which should indeed be imperceptible (we give an example of perturbed positive tokens in Figure 7 in the Appendix). We have added language to the caption clarifying this point for future readers.
> - The reviewer suggested that we should have designed our attack such that we add a perturbation to the input rather than to positive tokens. Our choice to investigate methods of perturbing positive tokens arose from the observation that perturbed tokens could potentially cause harm to many clean inputs, whereas a perturbed input would only cause misinterpretation of that specific input. For example, suppose that a particular imaging task consists of identifying whether an image contains melanoma. The user might want to be able to understand whether the model is making its decision based on asymmetry or mixed color of the proposed melanoma. It would be reasonable to collect a small set of positive tokens for each of these concepts and then use them repeatedly whenever one wants to interpret a model’s prediction based on these concepts. This persistent use of the same concept representatives over many input suggested that perturbation of these input had the potential to cause misinterpretation for many different input. We have added a few sentences to Section 3.1 making this motivation explicit.
> - We agree with the reviewer that more concept/class pairs would have made our work stronger. We added one more, scales/snake as noted above. This choice has the virtue that we can run a single concept against several ImageNet classes (specifically, all snake species). This can be found in Section 4 Figure 2.
> - The reviewer questioned the significance of the change in magnitude in Table 1. We updated Table 1 to report the direct TCAV score rather than the change in magnitude score, to more clearly show attack effectiveness.
> - The author brought up potentially adding more models to our transferability study. We agreed that this would make the paper stronger and therefore added MobileNetV2. This can be found in Section 5.1.
> - The reviewer requested to see a visualization after the attack on relative TCAV attack in Figure 6. We were unsure of the exact visualization requested; the perturbations for the PGD attack on relative TCAV are perceptually the same as those given in Figure 7. However, we also display an empirical DeepDream visualization for the CAV before and after a hidden layer attack in Figure 8. We qualitatively find that the visualizations for the CAV with the attacked concept set tend to resemble those of the CAV without the attack, albeit with unnatural hue and colors.

---

> > ### Comment · Reviewer_jcjC · 2021-11-27
> > **Thank you for your response**
> >
> > Thank you for your hard work on the rebuttal! The clarifications and additional results clarify some of my concerns. However, the updates in the revision were hard to distinguish due to a lack of distinction. For instance, I could not find the motivation provided in the response in Section 3.1 of the main paper. Further, I still have the following suggestions and questions:
> >
> > * As mentioned previously and also pointed out by Reviewer 6y3k, It is essential to evaluate recent concept-based methods [4,5] to demonstrate if the vulnerability of the concept-based explanations is a critical problem or a limitation of the standard TCAV method. Additionally, it is helpful to evaluate multiple datasets (e.g., OAI or CUB datasets have concept attributes), as mentioned in my original review.
> > * While I understand the motivation of the method, I believe it is critical to choose the adversary based on its strength. Hence, it would have been helpful to compare and discuss the vulnerability of the concept-based explanation to input-space perturbation compared to the proposed token pushing attack.
> > * The changes in Table 1 show a change in variance, as also highlighted by Reviewer oy63. Can the authors comment on the changes?
> >
> > ---
> > Overall, I feel the proposed method is interesting and novel;  however, I still believe the paper can be improved further with the above experimental evaluation, and another iteration of document revision and reviewing would be helpful.

---

> > > ### Author Response · Authors · 2021-11-29
> > > **Response to follow-up from reviewer jcjC**
> > >
> > > We thank you for your follow-up comments and your time. We include responses below, as well as highlight our changes to section 3.1.
> > >
> > > > - As mentioned previously and also pointed out by Reviewer 6y3k, It is essential to evaluate recent concept-based methods [4,5] to demonstrate if the vulnerability of the concept-based explanations is a critical problem or a limitation of the standard TCAV method. Additionally, it is helpful to evaluate multiple datasets (e.g., OAI or CUB datasets have concept attributes), as mentioned in my original review.
> > >
> > > We agree with the reviewer on the importance of evaluating the attack (and related attacks) on more concept-based interpretability methods and datasets. We believe that the effectiveness of the token pushing attack on faceted feature visualization (FFV) (which is significantly different from TCAV in many ways) offers some evidence that this attack is more general than just TCAV. In future iterations of this work, we plan to run experiments against additional models and datasets.
> > >
> > > > - While I understand the motivation of the method, I believe it is critical to choose the adversary based on its strength. Hence, it would have been helpful to compare and discuss the vulnerability of the concept-based explanation to input-space perturbation compared to the proposed token pushing attack.
> > >
> > > We thank the reviewer and agree that adding an assessment of input space perturbations would increase the coverage of threats to concept-based methods in the paper. We chose a vulnerability and threat profile we found particularly compelling. We include the additions to section 3.1 below for the convenience of the reviewer:
> > >
> > > >Actually, we show that TP attacks, which we propose below, work for F specific to both TCAV and FFV simultaneously by optimizing for an objective function that disrupts the fundamental mechanism by which TCAV, FFV, and other CBIMs work. As noted in the introduction, we centered our threat model around the positive tokens critical to CBIMs that, once perturbed, can cause persistent misinterpretation across numerous inputs. In contrast, a perturbation of an individual input image alone affects only the interpretation associated to that input.
> > >
> > > We plan to investigate and benchmark against input-space perturbations in future work.
> > >
> > > > - The changes in Table 1 show a change in variance, as also highlighted by Reviewer oy63. Can the authors comment on the changes?
> > >
> > > We include our response to Reviewer oy63 here on the change in effect and variance. We note (for clarity/other readers) that Table 1 now gives the TCAV magnitude scores, rather than the difference in TCAV magnitude scores between the baseline TCAV and the attacked TCAV as originally presented in Table 1.  With regard to the differences in the estimated effects, for our new experiments we not only made more TCAV runs with more negative concept examples, we also increased the D_k dataset size from 50 validation images to 800 images. While our new Table 1 results tend to be within the confidence intervals of the old results, we also suspect that the new dataset size is now more representative of the class under examination, and likely explains the change in the size of the effect.

---

### Official Review · Reviewer_6y3k · 2021-10-25

**Correctness:** 3
**Technical Novelty And Significance:** 2
**Empirical Novelty And Significance:** 2
**Recommendation:** 5
**Confidence:** 4

**Details Of Ethics Concerns:**

The paper does not discuss potential ethics concerns or impacts. I would highly suggest providing such a discussion because this paper is clearly within the security flag in the previous question. The authors could discuss from both the attack and defense perspectives.

**Main Review:**

This paper studies an important problem, i.e., the robustness of interpretation methods. Most of the existing works on adversarial robustness focus on deep learning models themselves. Litter attention has been drawn on interpretation methods. This paper works on this needed problem and proposes an interesting attack on existing interpretation methods. Despite providing some useful insights regarding discovering the vulnerabilities of interpretation methods, IMHO, this paper has not reached the bar of ICLR due to the following limitations.

1. The literature review of this paper is incomplete regarding both explanation methods and attacks on explanation methods. Regarding explanation methods, the paper fails to include perturbation-based/counterfactual-based white-box explanation methods (e.g., 1-3) and the large body of black-box explanation methods (e.g., 4-6). Regarding the attacks on interpretation methods, this paper misses the following works: 7-10, although they do not all target concept-based methods.

2. The main claim of this paper is not that accurate. This is not the first work that studies the robustness issue of concept-based explanation methods. 10 has discussed the robustness of concept-based against a type of attack.

3. Despite its correctness, the novelty and depth of the proposed technique are limited. Compared with the original PGD, the proposed method is just an application of PGD to a different type of classifier and input. IMHO, such a technical contribution does not reach the bar of a top-tier ML conference. This also raises another question: Why not choose other existing adversarial attack methods other than PGD. I would suggest the authors justify their design choice.

4. Regarding evaluation, this work lacks some potential comparison baselines and target methods. Regarding the target methods, the authors chose TCAV and another work that was recently published on distill. I would suggest the authors replace TCAV with more advanced methods with a similar core idea, i.e., 10-11. Regarding the comparison baselines, I would suggest the authors discuss and even evaluate why existing attacks pointed by me cannot be directly used, or applied through minor changes, to the concept-based methods targeted in this work. At least, the authors should discuss and compare with the attack mentioned in 10.

5. I would suggest the authors discuss and even evaluate potential countermeasures of the proposed attack.

6. The writing quality of this paper is limited. There is a branch of grammar errors and some sentences are hard to understand. I would suggest the authors run a grammar check and proofread the paper to ensure readability.


**Citations:**
1. Interpretable Explanations of Black Boxes by Meaningful Perturbation, ICCV 17.
2. Real time image saliency for black box classifiers, NeurIPS 17.
3. Explaining Image Classifiers by Counterfactual Generation, ICLR 19.
4. "Why Should I Trust You?": Explaining the Predictions of Any Classifier, KDD 16.
5. Explaining Deep Learning Models -- A Bayesian Non-parametric Approach, NeurIPS 18.
6. Learning to Explain: An Information-Theoretic Perspective on Model Interpretation, ICML 18.
7. Interpretable Deep Learning under fire, Usenix 20.
8. Fairwashing explanations with off-manifold detergent, ICML 20.
9. Robust and Stable Black Box Explanations, ICML 20.
10. Concept Bottleneck Models, ICML 20.
11. On Completeness-aware Concept-Based Explanations in Deep Neural Networks, NeurIPS 20.




**Summary Of The Paper:**

This paper proposes a novel adversarial attack against concept-based explanation methods. Technically, it utilizes a widely used attack method on DNN - PGD to generate perturbations. Adding these perturbations to concepts will mislead the target explanation methods to pinpoint meaningless parts in the input as important features.  Evaluation on a CNN model trained on the ImageNet dataset demonstrates the effectiveness of the proposed method.

**Summary Of The Review:**

IMHO, I am afraid this paper has not reached the bar of ICLR due to the following limitations. Detailed comments and suggestions can be found above.

1. The literature review of this paper is incomplete regarding both explanation methods and attacks on explanation methods.

2. The main claim of this paper is not that accurate. As discussed above, this is not the first work that studies the robustness issue of concept-based explanation methods. It has been studied in previous works.

3. The technical contribution of this paper is thin.

4. Regarding evaluation, this work lacks some potential comparison baselines.

5. Finally, I would suggest the authors discuss and even evaluate potential countermeasures of the proposed attack.

---

> ### Author Response · Authors · 2021-11-23
> **Response to 6y3k**
>
> We would like to start by thanking the reviewer for the thorough review of our paper and much helpful feedback. Where possible, we have tried to revise our work to address this feedback.
>
> - We have added most of the references suggested by the reviewer, including (1, 2, 3, 7, 8, 9, 10, 11). A few references we felt were not sufficiently related to justify adding.
>
> - When referencing [10] we assume that the reviewer is referring to the study of robustness of concept bottleneck models to changes in image background. While it is true that this experiment sits at the intersection of model robustness and concept-based interpretability methods, we do not agree that it represents an adversarial attack on a concept-based interpretability method. This is based on a few observations: (1) A fundamental component of adversarial attacks is that they are generally assumed to be subtle and relatively imperceptible to the human eye. Changes to image background are generally large and obvious. We see the experiment from [10] as part of the research program in robustness to distribution shift (for example Hendrycks, et. al. 2021) rather than adversarial robustness. (2) Related to the prior point, there is no realistic threat model associated with a background swapping attack. If an adversary can manipulate either training or deploy data to such an extent that they can change the image background (which we note also requires segmentation masks or laborious hand labeling), then the defender probably has much bigger problems. (3) Even if this study constitutes an attack, the target of the attack is model accuracy rather than the interpretability output. Part of the novelty of our work is that the attacks we investigate cause dramatic changes in the interpretability output (TCAV scores and feature visualizations). For these reasons, we do not believe that the experiment from [10] negates our claim to be the first work to look at adversarial attacks on concept-based interpretability methods.
>
> - The reviewer asked why we did not try other types of attacks (other than PGD) within our token pushing framework. We expect that similar results could be achieved using other attacks to perturb tokens with the effectiveness largely tied to the computational complexity of the method used (as is seen across the adversarial machine learning literature). We view the novelty of our method as arising from (a) the type of model we are attacking (concept-based interpretability methods), (2) the component of it we are attacking (positive tokens rather than input), and (3) the way we are attacking it (perturbing positive tokens to maximize shift of a centroid in a hidden layer). As such we chose to focus on these aspects in the work, leaving PGD as a generic placeholder attack that could be replaced by another in future work. We have added language to the beginning of Section 3 clarifying this.
>
> - The reviewer suggested that we replace TCAV with more advanced models introduced in follow-on work. This seems like a worthwhile goal that would be interesting to explore in future work. However, for this initial step into adversarial attacks on concept-based interpretability methods it seemed to us to be better to start with the simplest baseline which is already being used in applications. For example, we note that the paper “Acquisition of Chess Knowledge in AlphaZero” that was released this month uses CAV to probe AlphaZero. By applying our attack to the baseline we hope that our work will have wider applicability.
>
> - Following the reviewer’s advice, we reviewed the paper again and eliminated any grammatical errors that we came across.
>
> - Finally, noting that the reviewer flagged the paper for an ethics review, we added an ethics statement.
>
> Once again, we thank the reviewer for taking the time to give feedback for our paper.

---

### Official Review · Reviewer_oy63 · 2021-10-30

**Correctness:** 2
**Technical Novelty And Significance:** 3
**Empirical Novelty And Significance:** 2
**Recommendation:** 5
**Confidence:** 4

**Main Review:**

Strengths:

**Novelty**: to the best of my knowledge, the paper is the first adversarial attack for concept-based explanations.

**Clarity**: the paper is generally well-written and clear. I particularly appreciate that the authors are very clear about the goals of the method, and assumptions about the knowledge of the adversary.

-----
Weaknesses:

**Experiments**: Overall, I would have liked to see more rigorous experimentation.

Part of the experiments are well-designed -- e.g., the results in Figure 3, which demonstrate the change in feature visualization due to adversarial concepts is an interesting qualitative experiment.

However, some aspects of the method are not evaluated in the experiments.  For example, "Does the adversarial attack change the *relevance* of the concepts in the dataset?" This could be evaluated by investigating the change in the TCAV scores -- "are zebras no longer related to stripes after we attack the concept stripes" (assuming the model previously used this relationship)? The authors demonstrate that it changes the magnitude of the TCAV scores (in Figure 2). However, in the original TCAV paper, the sign of the score (rather than the magnitude is used). It's not directly clear to me unsure a larger magnitude directly implies a lower TCAV score.

The experiments are performed on a relatively small scale. The attack is evaluated on one dataset and predominantly on one model, InceptionV1 (although there is one experiment of the adversarial examples from InceptionV1 to ResNet-18). Further, the authors demonstrate the effect of the attack on two concepts (stripes and dots). I would have liked to see more datasets and more concepts.

*Significance of the results*: The authors have provided confidence intervals for their findings, which is great. However, often the confidence intervals are quite large, leading to statistical insignificance. Nevertheless, the authors state that "the attack targeting the layer mixed4b is successful across all layers considered", despite the large confidence intervals. Similarly, it's unclear whether the results in Figure 5 are significant (without seeing the error bars).  I'd like to encourage the authors to be more cautious in the conclusions they draw.


**Design/Method** : If I understand the method correctly, it assumes the adversarial examples are included before computing the concept vector $v_c^l$. However, this seems a bit strange for an adversarial attack -- normally, the model (including the interpretability method) is assumed to be fixed. I would've expected the method to focus on trying to change the explanation, e.g. the TCAV score, for fixed concept vectors. I would be curious to hear from the authors why the approach in the paper is more relevant and/or interesting than the aforementioned alternative?

-----
Minor comments:

Citations:
- The citation for adversarial examples (in section 3, final paragraph page 3) should be Szegedy et al., 2013 not Goodfellow et al., 2015.
- the discussion of some relevant work is missing -- see list below.


[1] Slack, Dylan, et al. "Fooling LIME and SHAP: Adversarial attacks on post hoc explanation methods." Proceedings of the AAAI/ACM Conference on AI, Ethics, and Society. 2020.

[2] Ghorbani, Amirata, Abubakar Abid, and James Zou. "Interpretation of neural networks is fragile." Proceedings of the AAAI Conference on Artificial Intelligence. Vol. 33. No. 01. 2019.

[3] Dombrowski, Ann-Kathrin, et al. "Explanations can be manipulated and geometry is to blame." arXiv preprint arXiv:1906.07983 (2019).

[4] Anders, Christopher, et al. "Fairwashing explanations with off-manifold detergent." International Conference on Machine Learning. PMLR, 2020.


**Summary Of The Paper:**

The authors derive a new adversarial attack, token pushing attack, that targets concept-based explanations. The adversarial perturbations are added to the concepts, resulting in a shift of the internal representation of a concept. The authors provide experiments that show that the adversarial attack changes the concept vectors and their visualization.

**Summary Of The Review:**

Overall, I'm recommending a borderline reject.
I think that the general concept of creating adversarial attacks for concept-based methods is interesting and relatively novel. However, the implementation of the idea requires assuming that the interpretability model will be re-trained, which limits the applicability of the method. Further, as a paper that introduces a new method, I think that more experimentation is required to demonstrate its effectiveness.

---

> ### Author Response · Authors · 2021-11-23
> **Response to oy63**
>
> We would first like to thank the reviewer for a thoughtful review. We tried to address as many of the points that the reviewer brought up in their “Weaknesses” section as we could. Those we did not address will be incorporated into our follow-on work:
>
> ### Experiments:
>
> - The reviewer questioned our choice of using TCAV magnitude scores instead of TCAV sign scores. We used the TCAV magnitude scores to have a more granular per-layer measure of concept significance with which to judge the effectiveness of the attack.
>
> - The reviewer pointed out that the scale of our experiments may have been too limited. Following the reviewer’s advice, we added an additional concept/class pair: scales/snake. One nice aspect of this addition is that we can use multiple ImageNet classes (in particular, different snake species from  ImageNet) when testing this concept. These additions can be found in Section 4, Figure 2.
>
> - The reviewer observed that the confidence intervals of our results did not always indicate statistical significance. To address this, we revisited our experiments, doing additional runs to iron out statistical irregularities, as well as adding confidence intervals to our plots. Our updated results appear in Table 1 and Figure 2.
>
>
> ### Design/Method:
>
> - When we initiated the present line of research, we considered the components of concept-based interpretability methods that could be perturbed to change the method’s outcome. The two obvious choices were the input (that datum that one wants to generate interpretations for) and the positive tokens that serve as representatives of a particular concept. We chose to focus on attacks on positive tokens for two reasons. (1) We imagined that in practice many input would be run against a few fixed sets of positive tokens. For example, suppose that a particular imaging task consists of identifying whether an image contains melanoma. The user might want to be able to understand whether the model is making its decision based on asymmetry or mixed color of the proposed melanoma. It would be reasonable to collect a small set of positive tokens for each of these concepts and then use them repeatedly whenever one wants to interpret a model’s prediction based on these concepts. This persistent use of the same concept representatives over many input suggested that perturbation of these input had the potential to cause misinterpretation for many different input. On the other hand, perturbing a single input would only change the interpretation of that particular input. Doubtless one can also come up with scenarios wherein changes to input result in more serious effects.
>
> - (cont.) (2) The second reason that we chose to focus on positive tokens is that this is a component that does not exist for other methods from explainable deep learning. As we note in our Related Works, there already exist methods of perturbing input such that attribution-based methods (such as saliency maps) produce misleading output. Thus we hoped that by targeting a novel component of concept-based interpretability methods we could make a larger contribution toward understanding the vulnerabilities of these models. We have added some sentences to Section 3.1 that try to make this choice more explicit.
>
> Again, we would like to thank the reviewer for their careful reading of our paper.

---

> > ### Comment · Reviewer_oy63 · 2021-11-25
> > **Follow-up**
> >
> > Thank you for your rebuttal; I appreciate the new experiments, as well as the further explanations. There are two aspects I'd like to discuss further.
> >
> > > "We used the TCAV magnitude scores to have a more granular per-layer measure of concept significance with which to judge the effectiveness of the attack."
> >
> > While I understand your point here, the original TCAV paper considers the sign rather than only the magnitudes. As such, I believe it would be more interesting to consider (or present in addition to your results), the change in the TCAV (sign) scores.
> >
> > >"We revisited our experiments, doing additional runs to iron out statistical irregularities, as well as adding confidence intervals to our plots"
> >
> > Thank you for running more experiments. I noticed that the magnitude of the values of changed -- specifically those in Table 1. While I would perhaps expect the variance to decrease with more samples, it's not clear to me why the estimated effects have changed this much. Could you perhaps comment a bit on the difference?

---

> > > ### Author Response · Authors · 2021-11-29
> > > **Response to follow-up from reviewer oy63**
> > >
> > > Thank you for your comments. We provide answers to some of your questions below.
> > >
> > > >While I understand your point here, the original TCAV paper considers the sign rather than only the magnitudes. As such, I believe it would be more interesting to consider (or present in addition to your results), the change in the TCAV (sign) scores.
> > >
> > > We agree that adding the sign score would add some value to our analysis and will include it in future versions of the paper. With that being said, we found that the effect of the attack measured via the TCAV sign scores tends to match that measured via the TCAV magnitude scores. We add the corresponding sign score table for the zebra/striped magnitude scores included in Table 1 in the paper.
> > >
> > >
> > > |                    | Layer TCAV sign score |               |               |               |
> > > | ------------------ | :-------------------: | :-----------: | :-----------: | :-----------: |
> > > | **Layer attacked** |      **mixed3a**      |  **mixed3b**  |  **mixed4a**  |  **mixed4b**  |
> > > | Baseline           |     0.63 +/- 0.01     | 0.80 +/- 0.01 | 0.59 +/- 0.02 | 0.57 +/- 0.03 |
> > > | *PGD attack on*    |                       |               |               |               |
> > > | mixed3a centroid   |     0.40 +/- 0.07     | 0.34 +/- 0.07 | 0.34 +/- 0.07 | 0.24 +/- 0.05 |
> > > | mixed3b centroid   |     0.41 +/- 0.07     | 0.37 +/- 0.07 | 0.38 +/- 0.07 | 0.39 +/- 0.07 |
> > > | mixed4a centroid   |     0.39 +/- 0.07     | 0.37 +/- 0.07 | 0.42 +/- 0.07 | 0.48 +/- 0.09 |
> > > | mixed4b centroid   |     0.37 +/- 0.07     | 0.36 +/- 0.07 | 0.4 +/- 0.07  | 0.45 +/- 0.08 |
> > > | mixed4c centroid   |     0.41 +/- 0.07     | 0.41 +/- 0.08 | 0.44 +/- 0.08 | 0.46 +/- 0.08 |
> > > | mixed4d centroid   |     0.4 +/- 0.07      | 0.42 +/- 0.08 | 0.4 +/- 0.07  | 0.45 +/- 0.08 |
> > >
> > >
> > > >Thank you for running more experiments. I noticed that the magnitude of the values of changed -- specifically those in Table 1. While I would perhaps expect the variance to decrease with more samples, it's not clear to me why the estimated effects have changed this much. Could you perhaps comment a bit on the difference?
> > >
> > > We note (for clarity/other readers) that Table 1 now gives the TCAV magnitude scores, rather than the difference in TCAV magnitude scores between the baseline TCAV and the attacked TCAV as originally presented in Table 1.  With regard to the differences in the estimated effects, for our new experiments we not only made more TCAV runs with more negative concept examples, we also increased the D_k dataset size from 50 validation images to 800 images. While our new Table 1 results tend to be within the confidence intervals of the old results, we also suspect that the new dataset size is now more representative of the class under examination, and likely explains the change in the size of the effect.

---

### Official Review · Reviewer_YeZA · 2021-11-03

**Correctness:** 2
**Technical Novelty And Significance:** 2
**Empirical Novelty And Significance:** 2
**Recommendation:** 3
**Confidence:** 4

**Main Review:**

While the paper focuses on an interesting problem, it does not seem to add anything new to our understanding of robustness of interpretability methods. As the paper itself mentions, the adversarial attacks on representations of the networks, and adversarial attacks on feature-based interpretability methods have already been out there. The only new thing that the paper seems to do is to apply the adversarial attacks on concept-based interpretability methods. This little amount of novelty, paired with the fact that the empirical results are not really surprising, means that the paper does not meet the bar for ICLR.

More in detail, are there any fundamental differences between the attacks on feature-based interpretability methods (e.g., [Dombrowski et al.](https://proceedings.neurips.cc/paper/2019/file/bb836c01cdc9120a9c984c525e4b1a4a-Paper.pdf)) and the proposed attack on the hidden representations? Do we expect one attack to be harder than the other? An empirical or a theoretical analysis (see Dombrowski et al, or Adebayo et al) would have been greatly helpful. Without such insights, the contributions of the paper seems limited.

**Summary Of The Paper:**

The paper proposes an adversarial attack against concept based interpretability methods. The key idea is to change the hidden layer representation at which the concept-based method is being applied in an adversarial manner. The results show that the proposed method is indeed able to fool TCAV and FFV methods.

**Summary Of The Review:**

Interesting idea, but lacks novelty.

---

> ### Author Response · Authors · 2021-11-23
> **Response to YeZA**
>
> We would like to start by thanking the reviewer for taking the time to read our paper and provide useful feedback.
>
> We would argue that there is perhaps more novelty to our approach than was summarized by the reviewer. From our perspective: (1) we identify a common component of several important concept-based interpretability methods (the use of linear probes in hidden layers), (2) we propose a method of perturbing this centroid that is specifically targeted at the way the centroid is used in concept-based interpretability methods, and (3) we assemble tools from adversarial machine learning to affect this attack. We are also the first (as far as we know) to propose a threat model for concept-based interpretability methods and the first attack on feature visualization. Based on the reviewer’s feedback, we have tried to summarize this novelty more explicitly at the end of the Introduction.
>
> The reviewer is correct that other works such as Dombrowski et. al. have already investigated how imperceptible changes to model input can radically change attribution. Critically though, both Dombrowski et. al. and Adebayo et. al. address attribution-based interpretability methods, not (the more general) concept-based interpretability methods. Concept methods either impose concept-priors in the hidden layers post-hoc (TCAV, FFV) or as regularization/prior during training (Concept Bottleneck models, Concept Whitening). These differ from the methods pointed out by the reviewer, in part, because the interpretability output can be far more diverse than the “explanation map” (considered by Dombrowski) output used for attribution. Our method specifically targets concept examples, rather than the input that is being interpreted, a component that is not present in attribution-based models. Hence it is unclear how attacks on attribution-based methods would transfer to concept-based models without significant modification. We have clarified the relationship of our work with prior work at the end of our Related Works section.
>
> We do appreciate the reviewer’s desire for a mathematical analysis of the brittleness of concept-based interpretability methods (as was performed in Dombrowski). We think this would be a useful direction and hope to make it the subject of future work.
>
> Once again, we thank the reviewer for taking the time to read and consider our work. We believe their feedback has helped to improve the presentation of our work.

---

### Author Response · Authors · 2021-11-23
**Introduction: rebuttal revision**

We would like to start by thanking all the reviewers for taking the time to read and give useful feedback on our paper. Where appropriate we have tried to address comments through additions and modifications to the paper. There were also several reviewer comments that we felt would be interesting follow-on work, but which we did not include in the current iteration. Below we summarize some of the major changes to the paper. Following that we provide responses to individual reviewers:

- More concept/class pairs: We received feedback from several reviewers who requested we run experiments against more concept/class pairs in our experimental section. We have since added scales/snake. One benefit of this pair is that there are multiple ``snake’’ classes in ImageNet corresponding to different species. This addition can be found in Section 4.

- Experimental runs and statistical significance: We increased the number of runs of many of our experiments to bolster the statistical significance of our results. Updated tables can be found in Section 4.

- Another model for transferability: At least one reviewer noted that our paper could have been made stronger if we included additional model architectures in our transferability study. We added MobileNetV2 to these experiments as can be found in Section 5.1.

- Ethics statement: Based on feedback from some of our reviewers we added an ethics statement.

---

### Decision · Program_Chairs · 2022-01-20

**Decision:**

Reject

**Comment:**

The work demonstrates that adversarially perturbing inputs can change the output of concept based explainability tools. Reviewers generally agreed that the writing was clear and the experiments were easily understood. Regarding novelty, reviewers noted that there are several existing works which study the adversarial robustness of explainability tools (one even has experiments specifically on concept based explainability tools). As a result, there is not much novelty in the finding that concept based explainability tools are sensitive to adversarial perturbation. Regarding the technical contribution of the algorithm, it is expected that standard optimization approaches (e.g. PGD) would be sufficient to break concept based explainability tools so there is not a clear technical challenge being solved in the work.

The work could be improved by refocusing the robustness analysis to derive new insights regarding the behavior of concept based explainability tools. In doing so, it would be beneficial to deemphasize the claims regarding novel security concerns---these methods don't even work reliably in non-adversarial settings, as evidenced by poor out-of-distribution robustness. It is expected that performance will be even worse under adversarial settings.